# Nuthatches vary their alarm calls based upon the source of the eavesdropped signals

Nora V Carlson [1,2,3]*, Erick Greene [4] & Christopher N Templeton[5]

Animal alarm calls can contain detailed information about a predator's threat, and heterospecific eavesdropping on these signals creates vast communication networks. While eavesdropping is common, this indirect public information is often less reliable than direct predator observations. Red-breasted nuthatches (*Sitta canadensis*) eavesdrop on chickadee mobbing calls and vary their behaviour depending on the threat encoded in those calls. Whether nuthatches propagate this indirect information in their own calls remains unknown. Here we test whether nuthatches propagate direct (high and low threat raptor vocalizations) or indirect (high and low threat chickadee mobbing calls) information about predators differently. When receiving direct information, nuthatches vary their mobbing calls to reflect the predator's threat. However, when nuthatches obtain indirect information, they produce calls with intermediate acoustic features, suggesting a more generic alarm signal. This suggests nuthatches are sensitive to the source and reliability of information and selectively propagate information in their own mobbing calls.

[1] Department of Collective Behaviour, University of Konstanz, Universitätsstraße 10, D-78457 Konstanz, Germany. [2] Centre for the Advanced Study of Collective Behaviour, University of Konstanz, Universitätsstraße 10, D-78457 Konstanz, Germany. [3] Department of Collective Behaviour, Max Plank Institute for Ornithology, Am Obstberg 1, 78315 Radolfzell am Bodensee, Germany. [4] Division of Biological Sciences and The Wildlife Biology Program, The University of Montana, Health Sciences 205, 32 Campus Drive, Missoula, MT 59812, USA. [5] Department of Biology, Pacific University, 2043 College Way, Forest Grove, OR 97116, USA. *email: nora.v.carlson@gmail.com

Predators are a major source of mortality for many species, and animals often use alarm calls to warn each other about danger[1–3]. Mobbing calls are a type of alarm call typically produced in response to predators that do not pose an immediate threat and function to recruit other individuals to drive these predators away from the area, making a future surprise attack less likely[4–7]. Variations in the acoustic structure of mobbing vocalizations can transmit detailed information about the predator[1,4,5,8,9], including the type, size, distance, behaviour, category, and threat level of the predator[1,6,10–19]. These calls are not only used by conspecifics, but are also eavesdropped on by heterospecifics as well[20–25]. Obtaining information about predators from these calls improves receivers' chances of survival[26,27] and their ability to respond appropriately to different types of predatory threats[18,28], and can also allow naïve individuals to learn about new predator species and the threats they pose[29–33]. Inappropriate responses to a predator threat can be costly either by overestimating the degree of threat leading to wasted time and energy that could be used for other important behaviours (e.g. foraging or searching for mates), or by underestimating the degree of threat leading to injury or death from the predator[34–36]. Thus, natural selection should favour alarm calling systems in which signals reliably reflect the threat level a potential predator poses and receivers assess the quality of the source of the information and base their responses on this information quality.

Although mobbing calls can encode detailed information about the danger posed by a given predator[5,16,24,37,38], the information that receivers extract from these calls may not provide the same degree of information that could be obtained through a direct encounter with the predator. This degree of reliability determines how useful indirect information is to an eavesdropper[20,39,40]. As individuals can vary their calls in response to the community or group composition[41–43], information obtained indirectly from alarm calls of others (i.e. public or indirect information) could sometimes provide an inaccurate representation of the threat[44–47]. A number of factors, for example life history features of a given species[43,48–54], or predator detection[55] could affect how individuals assess or respond to different threats; because the information included in these calls are thought to be a representation of the caller's perceived threat, not the threat itself[45], the reliability or accuracy of these calls to heterospecifics (and sometimes conspecifics) is potentially limited[45,53,56]. Although studies of alarm calls have provide a rich window on animal perception and behaviour, we still have important and outstanding questions about whether and how animals encode threat levels about predators depending on the source of the information—direct, personal information (e.g. hearing a predator calling) versus indirect, public information (e.g. hearing other species giving alarm calls)[44–46,54,56].

Here we experimentally test how eavesdroppers in a mixed-species bird community propagate information about predator threat in their alarm calls in response to direct, personal information versus indirect, public information. We study wild red-breasted nuthatches (*Sitta canadensis*), a species that frequently joins nuclear flocks of black-capped chickadees (*Poecile atricapillus*) during the winter. Chickadees are sensitive to the threat levels of different species of avian predators, and are known to vary their alarm calls when they encounter different size raptors[4] or overhear raptor calls[38]. Nuthatches are known to eavesdrop on their mobbing calls and respond appropriately to the information encoded therein[24,57], but little is known about whether or how nuthatches encode information about predator threat level in their own mobbing vocalizations.

Here we test the prediction that in response to direct information, nuthatches should produce finely-nuanced alarm calls that accurately reflect the level of threat of the predator; in

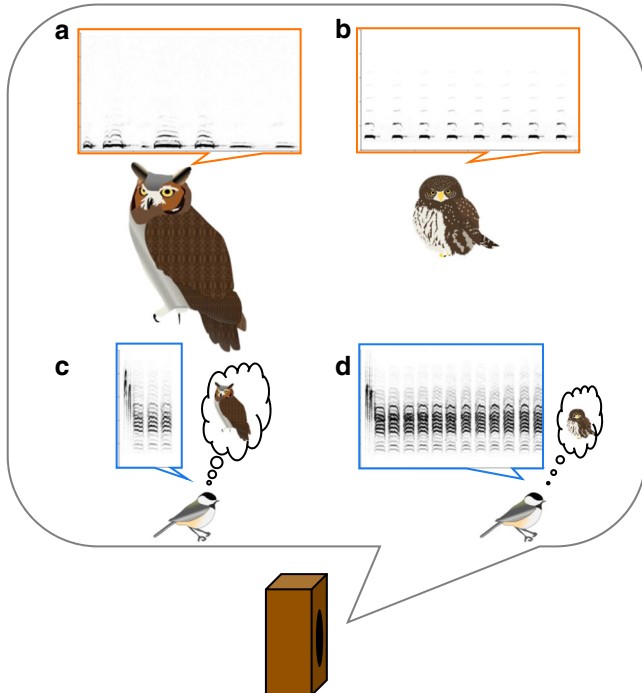

**Fig. 1 Schematic of playback experiment.** Nuthatches were exposed to playback of acoustic signals indicating a nearby predator of low or high threat. In the direct information treatment, we broadcast the vocalizations of **a** great-horned owls (low threat), and **b** northern pygmy-owls (high threat). In the indirect information treatment, we broadcast black-capped chickadee mobbing calls produced in response to **c** great horned owls, and **d** northern pygmy-owls.

contrast, in response to less reliable indirect public information, nuthatches should produce alarm calls that reflect the ambiguity in the source of information. In our study, we expose nuthatches to direct information indicating the presence of predators of varying threat levels (playbacks of high and low threat predator calls) and indirect information (chickadee mobbing calls in response to the same predators; Fig. 1). Nuthatches change acoustic features of their mobbing calls in response to increased levels of threat depending on the source of the information. They respond the least to low-threat predator calls, the most to high-threat predator calls, and intermediately to chickadee calls. They do not differentiate between indirect high- and low-threat calls. This suggests that (1) nuthatches only propagate reliable direct information but respond intermediately to less-reliable indirect information to avoid costs of miss-quantifying the degree of predator threat and (2) graded information in nuthatch calls is not a direct reflection of internal state.

## Results

**Acoustic features.** Nuthatches changed the acoustic features of their mobbing calls in response to different levels of threat based on the source of the information—direct or indirect—about predator threat. In response to calls of the high-threat northern pygmy-owl, nuthatches produced mobbing calls that had higher call rates, higher peak frequencies, and shorter call lengths compared to low-threat great-horned owls. In response to indirect public information about predators (mobbing calls of chickadees produced in response to these same owls), nuthatches produced mobbing calls with call rate, peak frequency, and call length parameters that were similar to both high- and low-threat chickadee calls, and intermediate compared with the low- and high-threat calls produced in response to direct information

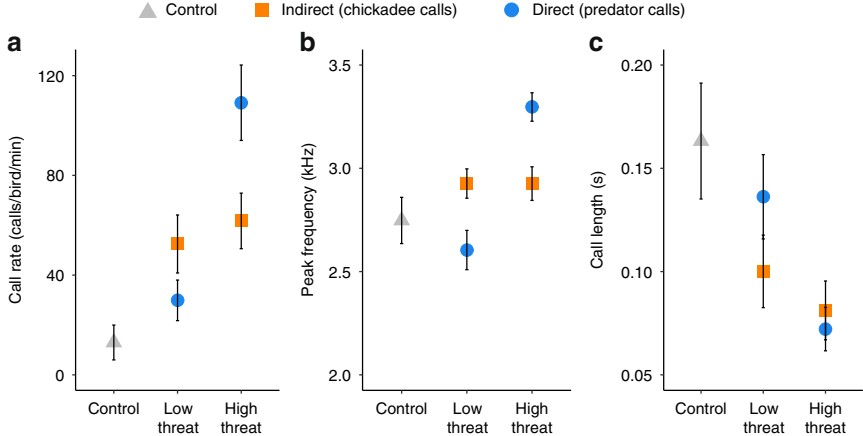

**Fig. 2 Acoustic features in response to predator threat and information source.** Acoustic parameters of nuthatch mobbing calls depend both on the threat level of the predator and the source of the information. Mean ± standard error of red-breasted nuthatch **a** call rate (calls/bird/minute), **b** peak frequency (kHz), and **c** call length (s) of mobbing calls produced in response to control (grey triangles), direct information (blue circles), and indirect information (orange squares). Source data are provided as Supplementary Data 1.

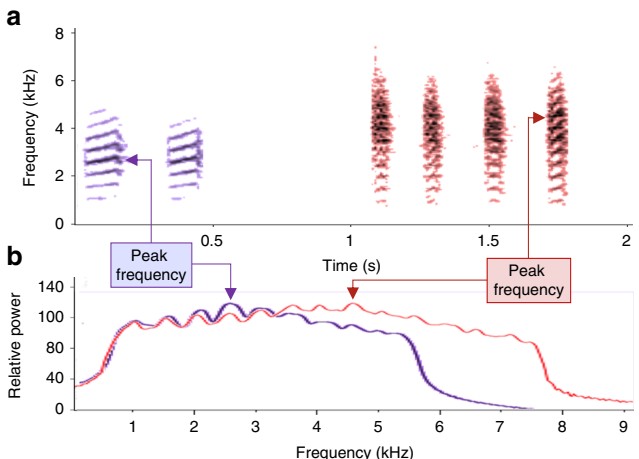

**Fig. 3 Visual representation of nuthatch and stimulus calls.** A spectrogram (**a**) and a power spectrum (**b**) of red-breasted nuthatch mobbing calls in response to low (left) and high (right) threat predators, illustrating the three acoustic parameters that varied with information source.

(Linear Mixed Model: call rate interaction: $\chi^2 = 6.43$, $P = 0.011$, peak frequency interaction: $\chi^2 = 12.78$, $P < 0.001$, call length interaction: $\chi^2 = 5.57$, $P = 0.018$; control: 38 sites, 632 calls, low-direct 19 sites, 1204 calls, high-direct: 21 sites, 5272 calls, low-indirect: 29 sites, 3172 calls, high-indirect 35 sites, 5205 calls; Figs. 2, 3; Supplementary Movie 1; Supplementary Table 1; Supplementary Data 1). There were no order:exemplar effects (Linear Mixed Model: call rate: $\chi^2 = 4.12$, $P = 0.846$; peak frequency: $\chi^2 = 3.30$, $P = 0.914$; call length: $\chi^2 = 10.42$, $P = 0.237$; Supplementary Table 1; Supplementary Data 1).

## Discussion

We found that red-breasted nuthatches produce mobbing calls in response to both direct and indirect information, but how they vary acoustic structures of these calls to transmit information about predator threat depends on the source of the information. In response to direct information (hearing predators), nuthatches produced very different alarm signals to each threat: hearing vocalizations of the very dangerous northern-pygmy owl elicited rapid, shorter, and higher pitched mobbing calls than calls elicited

from hearing vocalizations of the less dangerous great-horned owl. This is similar to other species that vary call rate, peak frequency, and call length in response to varying levels of predator threat[1,4,5,16]. Interestingly, the vocal behaviour of nuthatches hearing alarm calls of chickadees (indirect, public information) was quite different: despite clearly varying their investigatory and approach behaviour when eavesdropping on chickadee alarm calls representing different levels of threat[24], nuthatches did not in turn encode this information in their own vocalizations. These results indicate that nuthatches discriminate between direct and indirect, or public information, and this is reflected in the acoustic structure of their alarm calls.

Many animals use public information in uncertain or dangerous situations, including predator assessment and vigilance[54,58–64]. Templeton and Greene[24] showed that nuthatches obtain information about predator threat from chickadee mobbing calls; public information received from heterospecific mobbing calls could provide a relatively safe method for nuthatches to collect information about predators in their environment[20,54,65]. Nuthatches are potentially more vulnerable to predation than the species on which they are known to eavesdrop because of their restricted visibility associated with their foraging ecology (typically on or near tree trunks[66]) and tendency to spend more of the winter months in breeding pairs rather than larger groups[67]. These ecological factors, combined with previous research, suggests that nuthatches should use social information to learn about predators[24,54,57,68–70]. While nuthatches clearly do use indirect, public information to make behavioural decisions about mobbing, we found that they do not propagate this information in their own vocalizations. We suggest that the variation in vocal signalling behaviour in response to direct or indirect information could be due to differences in the reliability of these sources.

Public information is important for many reasons but is also generally not as reliable as personal/direct information[20,44–46,54,56,58,65,69,71,72]. Individuals—including the sender and receiver—can differ in their life history features, which can, in turn, affect their responses[20,51–54,70,73,74]. For example, Eastern chipmunks (*Tamias striatus*) not only respond to alarm calls with increased vigilance depending on their own boldness score (shyer individuals respond more), but also in response to the boldness score of the signaller (bolder individuals elicit increased vigilance)[51]. New Holland honeyeaters (*Phylidonyris novaehollandiae*) are slower to respond to aerial predators, and more likely to flee in response to an alarm call

when foraging on flowers than when perched[52]. When dealing with heterospecific alarm calls, these variables are further confounded by the fact that different species may also perceive the same predator as different threat levels[5]. For example, bird species that are more prevalent in the diet of a Eurasian pygmy-owl (*Glaucidum passerinum*) are more likely to mob that predator[74]. Additionally, heterospecifics, even allopatric species from the same family[5], may use different ways to encode predator threat information in their alarm calls[55]; and while some species can learn to recognize heterospecific calls either by associating a new call with their own alarm calls or by associating a new call with the presence of a predator, as superb fairy-wrens (*Malurus cyaneus*) do[32,33], not all species may be able to learn these associations.

The type of public information obtained can also greatly affect the reliability of the information[45]. Obtaining information by directly observing the state of an individual or the resources it is exploiting (i.e. how often an individual obtains food from a patch) is often highly reliable, but information obtained from the behavioural decisions of others (i.e. which patch an individual chooses) can be much less reliable[45]. Western Australian magpies (*Cractucus tibicen dorsalis*) for example, take a signaller's history of reliability (i.e. whether they previously signalled when a predator was present or absent) when making decisions about how to respond to an individual's alarm call[47] as individuals may differ in their reliability and therefore the threat signalled by the call. Chickadee calls reliably encode information about predator threat[44]. Nuthatches and chickadees share many predators, suggesting that they should also assess predator species similarly. However, while chickadee call types (i.e. mobbing vs. aerial alarm), like those of many species, reliably encode differences in type of threat and elicit correct search behaviour for different types of predators (i.e. looking up in response to aerial predator calls, and down in response to 'terrestrial' predator calls[16]), chickadee calls are likely not correlated directly with the threat itself but rather a measure of a particular chickadee's perceived threat level. It is likely that features other than predator species could impact a chickadee's overall threat assessment, including the predator's behaviour or distance, the presence of other conspecifics, or the calling individual's age, sex, experience, personality type, general stress level, internal state, or distance from protective cover[23,47–52,54,55,74–78]. Thus, individuals should be strategic in how they use different types of information[44,45,71–73]. As many species, including nuthatches, that are known to eavesdrop on mobbing calls have long-term pair bonds or kin nearby, it might be in an eavesdropper's best interest to produce reliable and accurate information to avoid eliciting inappropriate responses that could have negative fitness consequences for the receiver(s).

Due to the potential unreliability of public information obtained from eavesdropping on heterospecific alarm calls, nuthatches may avoid further propagating this information to their mate until they have assessed the situation and obtained more reliable, direct information themselves. Propagating incorrect information could cause a receiver to increase its risk of predation by under- or overestimating a threat. If a receiver underestimates a threat, it might be less vigilant, increasing its chance of predation. On the other hand, if a receiver overestimates a threat, it may increase conspicuous behaviour, like mobbing, which necessarily trades off with foraging and could even increase the likelihood of being depredated by other, unknown predators[79]. While nuthatches clearly use indirect public information to make behavioural decisions about predators, we found that the resulting alarm calls they produce reflect the uncertainty inherent in this less reliable source of information. Future research should use playback experiments to confirm that other nuthatches can decode information about predator threats and reliability from these acoustic variations in the nuthatch mobbing calls, and if they treat the indirect information in these calls similarly to the information in chickadee calls.

The finding that the mobbing vocalizations of nuthatches are not always directly correlated with their behavioural response adds an interesting piece of data to the debate about what type of information is reflected in variations of graded signals, i.e. where a given call type varies (e.g. number of repeated elements, peak frequency, etc.) across a gradient. Information coded in graded vocalizations, such as nuthatch and chickadee mobbing vocalizations, is often thought to be simply a reflection of the behavioural state of arousal of the calling bird[1], or level of risk[15]. Our findings, in conjunction with those of Templeton and Greene[24] and Billings et al.[55], suggest that this may not always be the case. Templeton and Greene[24] showed that nuthatches varied their anti-predator behaviour when they heard chickadee mobbing calls signifying different threat levels—implying heightened stress/arousal levels—while the current study demonstrates that these nuthatch behaviours do not necessarily translate to variation in their own vocalizations. Billings et al.[55] found that while Steller's jays do not change their foraging behaviour differently in response to visual compared to acoustic cues of predator presence, they did change graded aspects of their mobbing calls (i.e. duty cycle, number of elements)[55]. This suggests that while some mobbing calls use a graded acoustic structure, these gradations are not simply a reflection of an animal's internal state of arousal and could instead reflect more nuanced assessments of the information available in regard to predator encounters.

## Methods

**Subjects**. Red-breasted nuthatches tend to stay on year-round territories, but readily join mixed-species flocks with chickadees and other species[67]. Chickadees are gregarious, highly vocal, and readily mob predators[1,4,66].

**Stimuli**. To test if nuthatches encode information about predator threat in their alarm calls, we presented them with acoustic playbacks of predators that varied in threat level: low-threat (great horned owl, *Bubo virginianus*) and high-threat (Northern pygmy-owl, *Glaucidium gnoma*; Fig. 3). Small raptors, like the pygmy-owl, are more manoeuvrable and can accelerate faster than large raptors, and therefore pose a greater threat to small birds like nuthatches[80]. To test how nuthatches use different information sources, we also varied the type of information source: direct (calls produced by great-horned owls and northern pygmy-owls) and indirect (black-capped chickadee mobbing calls elicited in response to great-horned owls or northern-pygmy owls[4]; Fig. 1). We used vocalizations from two sympatric species commonly encountered in the winter—house sparrow (*Passer domesticus*) or Townsend's solitaire (*Myadestes townsendi*)—as control stimuli. The predator and control calls were obtained from the Macaulay Library of Cornell Lab of Ornithology (files: 87905, 50193, 119411, 59827(p), 50548, 22874b, 25635a, 11117a, 140258). To reduce pseudoreplication, we had a library of three different examples for each stimulus. Low threat chickadee calls had a call rate of 21.75 ± 3.63 calls/min, and 3.45 ± 0.34 D elements/call and high threat chickadee calls had a call rate of 27.25 ± 2.23 calls/min, and 7.30 ± 1.75 D elements/call.

**Playback trials**. We conducted auditory playback to pairs of nuthatches at 60 locations in and around Missoula, Montana (46°, 50′ N; 114°, 02′ W), during the winters of 2005 (ref. [24]), and 2012, 2013, and 2015, and in three locations in and around Mazama, Washington (48°, 35′ N; 120°, 24′ W), during the winter of 2016. We used a repeated measures experimental design where individuals at each location were presented with each of the five playback stimuli (with the exception of the 2007 data used from ref. [24], in which birds only received chickadee call and control playbacks). While we could not be sure of individual identity for all trials as individuals were not colour ringed, nuthatches remain territorial in the winter[67], study sites were at least 500 m apart, and the majority of the sites were not reused across study years. During each playback trial, we located nuthatches that were not in a mixed species flock in order to isolate the responses of the nuthatches to the playbacks without the actual presence and behaviour of chickadees or other species influencing their behaviour. We then played back one stimulus for 1 min using a Pignose Legendary 7-100 speaker (frequency response curve is relatively flat between 500 Hz and 17,000 Hz, Pignose, Las Vegas, NV, USA). This equipment produces high-fidelity non-frequency warped playback characteristics in the hearing range of nuthatches[81] and is commonly used in playback experiments to birds (e.g. ref. [82]). We calibrated the peak amplitude of each playback stimulus to a natural volume of 75 dB SPL (A-weighting) measured 1 m from the speaker, which

observers judged by ear to be similar to natural call amplitudes. For each focal group of nuthatches, the order of presentation of the stimuli and the exemplar for each stimulus were randomized.

The acoustic responses of the nuthatches were recorded with a Sennheiser ME 67 shotgun microphone (Sennheiser, Wedemark, Germany) onto a SONY TCM-5000 tape recorder or a Marantz PMD661 digital recorded (48 kHz sampling rate and 24-bit depth). Due to variation in sampling effort, sample sizes varied somewhat across treatments: control ($n = 38$), low-threat direct information ($n = 19$), low-threat indirect information ($n = 29$), high-threat indirect information ($n = 35$), and high-threat direct information ($n = 21$).

**Ethical note**. We simulated the presence of raptors by playing their calls to wild, free-living birds. We limited our experiments to periods between 1 h after sunrise and 1 h before sunset to allow birds to feed after waking up and before the night. Our experiments conformed to the standards outlined in the ASAB/ABS Guidelines for the Use of Animals in Research and were conducted under the University of Montana's Institutional Animal Care and Use Committee's Animal Use Protocols (ACC 022-01, 049-14EGDBS-080814, 001-11EGDBS-080511).

**Acoustic analysis**. We analysed the nuthatch vocalizations using Raven acoustical software (RavenPro 1.4 & 1.5; Bioacoustics Research Program 2011) with a spectrogram frequency grid resolution of 23.04 Hz, and a Hann window function. Nuthatches varied in their latency to arrive and begin mobbing after the playback, likely due to variation in distance from the speaker or behaviour at the start of the playback trial. Because of this variation, we standardized our measure of mobbing response by focusing the analyses by using the 60 s after the birds arrived and began intensively mobbing. We recorded (1) the number of calls produced, (2) peak frequency: the frequency (kHz) at which the most power occurred, and (3) call length (seconds; Fig. 3). We chose these acoustic features as they are commonly used by other species, including chickadees and other species of Paridae, to encode information about predator threat in graded signals[1,4,5]. Occasionally nuthatches would respond to live raptors that flew by just before or during the experiments; we excluded these trials from the analyses. Rarely nuthatches responded by approaching the playback speaker without vocalizing themselves; these trials were also excluded.

**Statistical analysis**. To test how nuthatches varied their calling behaviour in response to different predators and sources of information, we tested for an interaction between predator threat and information source. We have observed that nuthatches vary in the time they take to begin mobbing after playbacks, escalate from silence to a steady mobbing rate, and vary in how long they respond. Because of this variation, we standardized our measure of mobbing response by focusing the analysis on the 60 s with the highest call rate (calls/s) during mobbing response for each trial. We used R (v3.6.1) statistical software[83,84] and the lme4 package to perform data analyses. We generated linear mixed models with a Gaussian distribution to test for the interaction between predator threat and information type on red-breasted nuthatch (1) call rate (total calls per minute/ number of individuals present), (2) peak frequency (kHz; the frequency at which the most power occurs), and (3) call length (seconds). Our models included the fixed factors of predator threat (three levels: control, low threat/great-horned owl, high threat/northern pygmy-owl), information type (three levels: control, direct, and indirect), the number of nuthatches present, call exemplar, presentation order, and the interaction between call exemplar and presentation order. Year and location were included in the models as random effects in order to limit the effects of pseudoreplicaton due to multiple individuals and calls being produced at each location. Nuthatches responded similarly to both species of control playback (house sparrow and Townsend's solitaire), so these were pooled for further data analysis. We included call exemplar to control for playback exemplars, presentation order to control for order effects, and the interaction between exemplar and order to control for specific combinations thereof (i.e. control 1 being first), and number of red-breasted nuthatches to control for differences in number of nuthatches ((mean ± SEM): 2.317 ± 0.11 nuthatches; the majority of the trials had 1–4 nuthatches present, but in 7% of trials there were 6 or 8 individuals present). We observed a potential exemplar–order interaction in the data; however, this was likely due to stochasticity in which exemplars were used during the relatively small sample size of control trials where we recorded vocalizations. To determine if there were significant factors or an interaction between predator threat and information type, we compared models using likelihood-ratio tests (using the drop1 function). All $P$ values we report are two tailed.

**Reporting summary**. Further information on research design is available in the Nature Research Reporting Summary linked to this article.

## Data availability
Data used in this study can be found in Supplementary Data 1.

## Code availability
R Code used in this article for analysis can be found in Supplementary Software 1.

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

## Acknowledgements

We are grateful to Luke Rendell and Lauren Chan for commenting on an earlier version of this manuscript; the many individuals who provided access to land and bird feeders, and to Dusty Thomas, Alexis Billings, Devin Landry, Madison Furlong, Ian Anderson, and Samuel Case who helped with fieldwork. Special thanks to Mark Reiling, Libby Maclay and Sapphire Ranch. This research was supported by the Philip L Wright Memorial Research Award and the Montana Integrated Learning Experience for Students (to N.V.C.); NERC fellowship (NE/J018694/1), National Institutes of Health Auditory Neuroscience Training Grant, funds from Pacific University, and the MJ

Murdock Charitable Trust (to C.N.T.); and The National Science Foundation DEB 1258003 (to E.G. and Chris Clark).

## Author contributions

N.V.C., E.G. and C.N.T. developed and designed the experiments, carried out field work, and helped write and edit the manuscript. N.V.C. analysed the acoustic data and ran all statistical tests.

## Competing interests

The authors declare no competing interests.
