## [Peer Review File · Nature Communications]

Reviewers' Comments:

Reviewer #1:

Remarks to the Author:

The paper represents a thorough and interesting aspect of heterospecific communications and graded responses according to information source. In most ways, I cannot fault the paper either in method or clarity of results. It is also carefully written and the paper requires very few corrections. For instance, very minor corrections are:

Line 60 spelling error- very/vary: are known to vary their alarm calls; Line 192/3 double brackets- remove one (preferably change bracket for comma in line 192) Line 482/3 incomplete citation --pls complete-(my copy says: Bates, D., Maechler, M., Bolker, B. and Walker, S., 2014. lme4: Linear mixed-effects models using Eigen and S4. R package version, 1(7), pp.1-23.)

However, the paper suffers from a dated perspective and does not engage with recent and relevant literature in the field, not even with publications that have recently appeared in Nature communication.

It seems that the paper stops in its own theoretical framework in 2014, except for inclusion of two titles that are later, and does not engage with recent findings. A period of five years is a long time in science, particularly in a field that is of continued interest and vibrant. I am thinking of Suzuki's work et al. (2016, 2018) Magrath et al. -2015-2018, and Potvin et al. (2018) and various others.

The literature has been quite rich and innovative in the field in which the authors present their findings and I firmly believe that it is essential that a paper is fully engaged in (or delineated from) current debates in the field and that such engagement is apparent in the introduction, possibly in the method and most definitely in the discussion. The paper consequently is not within a framework of current ideas and research findings.

As a reader, one needs to be convinced that the paper is fresh and has something new to say that has not already been described or discovered or has been debated in the recent literature. If this is an old paper, then it needs to be updated (datedness also noticeable in the programs used in Raven and the. lme4: Linear mixed-effects models package- both have more recent versions). If it is a paper that has purposely omitted latest research findings, for whatever reason, one can only advise that this strategy does not work and would appear to me to run counter to the very raison d'être of research reporting. At the very least, it leaves the reader dissatisfied.

Reviewer #2:

Remarks to the Author:

Review of NCOMMS-19-23668

I have reviewed the paper "Nuthatches vary their alarm calls based upon the reliability of the eavesdropped signals".

This study focuses on the influence of direct vs. indirect information sources on the further propagation of predator-specific information in the alarm calls of red-breasted nuthatches. Through the implementation of a controlled playback experiment the authors show that when receiving direct information (i.e. predator vocalisations), red-breasted nuthatches reliably vary their alarm call structure to accurately communicate this threat. However, when receiving indirect information (i.e. through the alarm calls of chickadees responding to the same predators) they

produce alarm calls that are more intermediate or “general” in structure.

I thought the paper was very well written and the results interesting. Of particular relevance is the fact that the results speak against the popular theory in animal communication that information encoded in graded vocalisations varies systematically with arousal: although exposed to likely highly arousing chickadee alarm calls, this did not result in nuthatcher vocalisations of comparable structure.

I think this paper will be of interest to researchers from a broad spectrum of disciplines, from animal communication, to behavioral ecology and comparative psychology. I have a few comments that the authors might want to consider when revising the MS.

Specific comments:

1. L82: Perhaps the authors could spell out the interaction effects more, which I feel would help the reader interpret the results more clearly.

2. L84: From the results it seems that for call length there is not an interaction between predator threat and the source of information but it is generally discussed as if there is. I think it would be good for the authors to note that the interaction applied for 2 of the 3 acoustic variables measured.

3. L96: Isn't this the same result as on L82? It is not clear why the authors bring this up a second time rather than just discussing this is a trend when they first describe the results (L82).

4. L210-214: This reads more like introduction than methods.

5. L250-251: It is not really clear what the authors mean by “good playback characteristics” – please elaborate.

L258: If the authors were not able to identify individuals, is it possible that the same individual contributed more than one call to the acoustic response data set? Is this problematic in terms of pseudoreplication? In line with this, can the authors add sample sizes of calls analysed to the results section (i.e. L81-99).

6. L318: Wouldn't Likelihood-ratio tests not have been more appropriate here?

Minor points:

L60: change “very” to “vary”.

L293: Change to : “vary in the time they take to begin mobbing...”

Reviewers' Comments:

Reviewer #1 (Remarks to the Author):

The paper represents a thorough and interesting aspect of heterospecific communications and graded responses according to information source. In most ways, I cannot fault the paper either in method or clarity of results. It is also carefully written and the paper requires very few corrections. For instance, very minor corrections are:

Line 60 spelling error- very/vary: are known to vary their alarm calls;
We have fixed this typo (line 65).

Line 192/3 double brackets-remove one (preferably change bracket for comma in line 192)
We have changed one of the brackets for a comma in line 192 as suggested. (lines 196 & 198)

Line 482/3 incomplete citation --pls complete-(my copy says: Bates, D., Maechler, M., Bolker, B. and Walker, S., 2014. lme4: Linear mixed-effects models using Eigen and S4. R package version, 1(7), pp.1-23.)
We have updated the citation so it is correct (line 545-546).

However, the paper suffers from a dated perspective and does not engage with recent and relevant literature in the field, not even with publications that have recently appeared in Nature communication.

It seems that the paper stops in its own theoretical framework in 2014, except for inclusion of two titles that are later, and does not engage with recent findings. A period of five years is a long time in science, particularly in a field that is of continued interest and vibrant. I am thinking of Suzuki's work et al. (2016, 2018) Magrath et al. -2015-2018, and Potvin et al. (2018) and various others.

The literature has been quite rich and innovative in the field in which the authors present their findings and I firmly believe that it is essential that a paper is fully engaged in (or delineated from) current debates in the field and that such engagement is apparent in the introduction, possibly in the method and most definitely in the discussion. The paper consequently is not within a framework of current ideas and research findings.

As a reader, one needs to be convinced that the paper is fresh and has something new to say that has not already been described or discovered or has been debated in the recent literature. If this is an old paper, then it needs to be updated

The reviewer is correct that this was a manuscript we initially drafted several years ago. We have carefully updated the references. In the revised manuscript, we have included the suggested citations and several more. This has allowed us to put our results in a more current framework.

Here are more recent references that we now include:

- Billings, A. C., Greene, E., & MacArthur-Waltz, D. (2017). Steller's jays assess and communicate about predator risk using detection cues and identity. *Behavioral Ecology*, 1–8. <http://doi.org/10.1093/beheco/ax035> (lines 425-427)
- Couchoux, C., Clermont, J., Garant, D. & Réale, D. Signaler and receiver boldness influence response to alarm calls in eastern chipmunks. *Behav Ecol* **29**, 212–220 (2017). (lines 461-463)
- Cunningham, S., & Magrath, R. D. (2017). Functionally referential alarm calls in noisy miners communicate about predator behaviour. *Animal Behaviour*, 129, 171–179. <http://doi.org/10.1016/j.anbehav.2017.05.021> (lines 368-369)
- Dutour, M., Lena, J.-P., & Lengagne, T. (2017). Mobbing behaviour in a passerine community increases with prevalence in predator diet. *Ibis*, 1–7. <http://doi.org/10.1111/ibi.12461> (lines 518-520)
- Griesser, M., & Suzuki, T. N. (2017). Naive juveniles are more likely to become breeders after witnessing predator mobbing. *The American Naturalist*, 189(1), 58–66. <http://doi.org/10.1086/689477> (lines 401-402).
- Igic, B., Ratnayake, C. P., Radford, A. N., & Magrath, R. D. (2019). Eavesdropping magpies respond to the number of heterospecifics giving alarm calls but not the number of species calling. *Animal Behaviour*, 148, 133–143. <http://doi.org/10.1016/j.anbehav.2018.12.012> (lines 464-466)
- Kalb, N., Anger, F., & Randler, C. (2019). Subtle variations in mobbing calls are predator-specific in great tits (*Parus major*). *Scientific Reports*, 1- 7. <http://doi.org/10.1038/s41598-019-43087-9> (lines 372-374).
- Kalb, N. & Randler, C. Behavioral responses to conspecific mobbing calls are predator-specific in great tits (*Parus major*). *Ecol Evol* **9**, 9207–9213 (2019). (lines 370-371)
- Lilly, M. V., Lucore, E. C., & Tarvin, K. A. (2019). Eavesdropping grey squirrels infer safety from bird chatter. *PLoS ONE*, 14(9), e0221279–15. <http://doi.org/10.1371/journal.pone.0221279> (lines 496-497)
- Magrath, R. D., Haff, T. M., McLachlan, J. R. & Igic, B. Wild birds learn to eavesdrop on heterospecific alarm calls. *Curr Biol* **25**, 2047–2050 (2015). (lines 415-416).
- McIvor, G. E., Lee, V. E. & Thornton, A. Testing social learning of anti-predator responses in juvenile jackdaws: the importance of accounting for levels of agitation. *R Soc Open Sci* **5**, 171571–12 (2018). (lines 467-469)
- McLachlan, J. R., Ratnayake, C. P., & Magrath, R. D. (2019). Personal information about danger trumps social information from avian alarm calls. *Proceedings of the Royal Society B: Biological Sciences*, 286(1899), 20182945–9. <http://doi.org/10.1098/rspb.2018.2945> (lines 470-472).

Pell, F. S. E. D., Potvin, D. A., Ratnayake, C. P., Fernández-Juricic, E., Magrath, R. D., & Radford, A. N. (2018). Birds orient their heads appropriately in response to functionally referential alarm calls of heterospecifics. *Animal Behaviour*, 140, 109–118. (lines 375-377)

Potvin, D. A., Ratnayake, C. P., Radford, A. N., & Magrath, R. D. (2018). Birds learn socially to recognize heterospecific alarm calls by acoustic association. *Current Biology*, 28(16), 2632-2637. (lines 412-414).

Silvestri, A., Morgan, K. & Ridley, A. R. The association between evidence of a predator threat and responsiveness to alarm calls in Western Australian magpies (*Cracticus tibicen dorsalis*). *PeerJ* 7, e7572–17 (2019). (lines 450-452)

Suzuki, T. N., Wheatcroft, D., & Griesser, M. (2016). Experimental evidence for compositional syntax in bird calls. *Nature communications*, 7, 10986. (lines 347-348)

Suzuki, T. N. (2018). Alarm calls evoke a visual search image of a predator in birds. *Proceedings of the National Academy of Sciences*, 115(7), 1541-1545. (lines 378-379)

Wheeler, B. C., Fahy, M., & Tiddi, B. (2019). Experimental evidence for heterospecific alarm signal recognition via associative learning in wild capuchin monkeys. *Animal Cognition*, 1–9. <http://doi.org/10.1007/s10071-019-01264-3> (lines 409-411).

. . . the datedness also noticeable in the programs used in Raven and the. lme4: Linear mixed-effects models package- both have more recent versions. If it is a paper that has purposely omitted latest research findings, for whatever reason, one can only advise that this strategy does not work and would appear to me to run counter to the very raison d'être of research reporting. At the very least, it leaves the reader dissatisfied.

We have reanalyzed the data using a newer version of the lme4 R package for the revised manuscript.

As far as the data analysis is concerned, while we realize that one of the Raven versions used was not the most recent version we believe that there are no significant differences in calculations for length or peak frequency between the two versions, as the software updates have been in other areas. As for the lme4 version used for modeling the data, we have re-run the statistics on the more recent version and included this in our revision (line 299).

Reviewer #2 (Remarks to the Author):

Review of NCOMMS-19-23668

I have reviewed the paper “Nuthatches vary their alarm calls based upon the reliability of the eavesdropped signals”.

This study focuses on the influence of direct vs. indirect information sources on the further propagation of predator-specific information in the alarm calls of red-breasted nuthatches. Through the implementation of a controlled playback experiment the authors show that when receiving direct information (i.e. predator vocalisations), red-breasted nuthatches reliably vary their alarm call structure to accurately communicate this threat. However, when receiving indirect information (i.e. through the alarm calls of chickadees responding to the same predators) they produce alarm calls that are more intermediate or “general” in structure.

I thought the paper was very well written and the results interesting. Of particular relevance is the fact that the results speak against the popular theory in animal communication that information encoded in graded vocalisations varies systematically with arousal: although exposed to likely highly arousing chickadee alarm calls, this did not result in nuthatcher vocalisations of comparable structure.

I think this paper will be of interest to researchers from a broad spectrum of disciplines, from animal communication, to behavioral ecology and comparative psychology. I have a few comments that the authors might want to consider when revising the MS.

Specific comments:

1. L82: Perhaps the authors could spell out the interaction effects more, which I feel would help the reader interpret the results more clearly.

We have elaborated on the nature of the interaction (lines 91-98)

2. L84: From the results it seems that for call length there is not an interaction between predator threat and the source of information but it is generally discussed as if there is. I think it would be good for the authors to note that the interaction applied for 2 of the 3 acoustic variables measured.

We have clarified which call features were significant, though with the reviewer’s suggested likelihood-ratio tests rather than the Wald Chi squared tests, the call length changed to fall under the 0.05 line of statistical significance. (lines 93-98)

3. L96: Isn’t this the same result as on L82? It is not clear why the authors bring this up a second time rather than just discussing this is a trend when they first describe the results (L82).

Our original intention was to explain that there was an interaction and then explain in detail how the nuthatches responded for each metric. However, we have consolidated this section into one paragraph to avoid repeating ourselves (lines 90-98)

4. L210-214: This reads more like introduction than methods.

Thank you for pointing this out, we have moved this section to the introduction (lines 67-69) and removed it from the methods (line 215).

5. L250-251: It is not really clear what the authors mean by “good playback characteristics” – please elaborate.

We have clarified what we mean by good here (lines 251-252)

L258: If the authors were not able to identify individuals, is it possible that the same individual

contributed more than one call to the acoustic response data set? Is this problematic in terms of pseudoreplication? In line with this, can the authors add sample sizes of calls analyzed to the results section (i.e. L81-99).

We included site in the random effects specifically for this reason. By treating each site as an 'individual' we have tried to account for pseudoreplication due to individuals producing multiple calls. We have added the number of calls to the results section (lines 98-101)– though our sample size was treated as the number of sites, not individual calls. Additionally, the call rate is the total number of calls divided by the number of individuals present to get a calls/individual call rate for each site. We have also tried to clarify this in the methods (lines 308-310)

6. L318: Wouldn't Likelihood-ratio tests not have been more appropriate here?

Thank you for your suggestion. We have re-run the analysis the model using likelihood-ratio tests and have updated the text accordingly (lines 321-322).

Minor points:

L60: change "very" to "vary".

Thank you for pointing this out, we have fixed this typo (line 65).

L293: Change to : "vary in the time they take to begin mobbing..."

Thank you for pointing this out, we have corrected this (line 295-296).

REVIEWERS' COMMENTS:

Reviewer #1 (Remarks to the Author):

I am satisfied that the corrections are completed and, accordingly, this paper would now appear to be publishable

Reviewer #2 (Remarks to the Author):

Review of: NCOMMS-19-23668A

I have re-reviewed the paper "Nuthatches vary their alarm calls based upon the reliability of the eavesdropped signals".

I think the authors have done a good job at addressing the issues I raised. I do, however, just have a few more final changes that would be important to incorporate before publication.

L78-81: This is direct repetition of L70-73.

L95-96: It is not clear how parameters that differ in their dimensionality (time, frequency) can be considered as "similar to each other". Can the authors clarify here?

L91-98: The authors have definitely expanded on the interaction effect, but I was thinking this could be stated more clearly and generally, specifically, that the effect that threat level had on an acoustic variable differed based on whether the information was direct or indirect. They can then go on to report the exact results of the mixed effects models.

L182: depredation? Why not just predation?

L310: remove additional bracket.

L317-318: Maybe change to "We observed a potential exemplar-order interaction in the data.."

L329: Should that be "Luke"?

Responses to Reviewers by Carlson, Greene and Templeton

Reviewer's Comments:

Reviewer #1 (Remarks to the Author):

I am satisfied that the corrections are completed and, accordingly, this paper would now appear to be publishable

Thank you, we are glad we were able to address all your concerns.

Reviewer #2 (Remarks to the Author):

Review of: NCOMMS-19-23668A

I have re-reviewed the paper "Nuthatches vary their alarm calls based upon the reliability of the eavesdropped signals".

I think the authors have done a good job at addressing the issues I raised. I do, however, just have a few more final changes that would be important to incorporate before publication.

Thank you, we are glad that we were able to address most of your concerns and have revised our manuscript further regarding the below suggestions.

L78-81: This is direct repetition of L70-73.

Thank you for pointing that out, we have removed the repetition. (line 121)

L95-96: It is not clear how parameters that differ in their dimensionality (time, frequency) can be considered as "similar to each other". Can the authors clarify here?

Here we meant that the response to high and low threat indirect information were similar to one another, rather than the specific time/frequency parameters. We have clarified our meaning in the text (line 149)

L91-98: The authors have definitely expanded on the interaction effect, but I was thinking this could be stated more clearly and generally, specifically, that the effect that threat level had on an acoustic variable differed based on whether the information was direct or indirect. They can then go on to report the exact results of the mixed effects models.

We have added in a more general first sentence explaining this difference. (lines 142-144)

L182: depredation? Why not just predation?

We have replaced depredation with predation instead (line 311)

L310: remove additional bracket.

We have removed the additional bracket as suggested (line 467)

L317-318: Maybe change to "We observed a potential exemplar-order interaction in the data.."

We have revised the sentence as suggested (line 474)

L329: Should that be "Luke"?

It should be Luke, thank you for catching that (line 717).